# Parous rate and longevity of anophelines mosquitoes in bure district, northwestern Ethiopia

**Tilahun Adugna** [1¤]*, **Emana Getu**[2], **Delenasaw Yewhelew**[3]

**1** Department of Biology, Faculty of Natural and Computational Sciences, Debre Tabor, Amhara, Ethiopia,
**2** Department of Zoological Science, Addis Ababa University, Addis Ababa, Addis Ababa, Ethiopia,
**3** Department of Medical Laboratory Sciences and Pathology, College of Health Sciences, Jimma, Oromia, Ethiopia

¤ Current address: Department of Biology, Debre Tabor University, Debre Tabor, Ethiopia
* tilahun46@yahoo.com, adugnawassie@gmail.com

## Abstract

The intensity of malaria transmission is measured by parous rate, daily survival rate, human blood meal frequency, sporozoite rate, and entomological inoculation rates. Female parous status is a key index of vector competence, adult vector longevity, recruitment rate of adult, and the length of a gonotrophic cycle. Hence, the present study was aimed to investigate the parous rate and the longevity of Anopheles mosquitoes in Bure District, Northwestern Ethiopia. Parous rate was estimated as the number of mosquitoes with parous ovaries divided by the number of females dissected multiplied by 100. Mosquito life expectancy (longevity as d) was estimated by. One way- ANOVA was applied to confirm the presence of parous rate difference in the villages ($p < 0.05$). A total of 952 unfed hosts-seeking Anopheles mosquitoes was dissected for parous rate determination. The overall parous rate of An. arabiensis in the district was 52.0%, and the highest parous rate was recorded in Shnebekuma than other villages ($F_{2, 33} = 6.974$; $p = 0.003$). Similarly, the parous rate of An. cinereus showed significant variation among villages ($F_{2, 33} = 5.044$, $p = 0.012$) and the highest rate (63.0%) was recorded in Bukta. The mean longevity of An. funestus, An. arabiensis, An. coustani, An. squamosus, An. pharoensis, and An. cinereus was 6.5 days, 4.6 days, 3.5 days, 3.7 days, 2.7 days, and 2.2 days, respectively. The longevity of each species was not sufficient to complete the life cycle of malaria parasite for malaria transmission throughout the year because P. falciparum requires from 12–14 day.

**Data Availability Statement:** All relevant data are within the manuscript and its Supporting Information files.

**Funding:** We didn't get any research grant (funder) to this research work. Addis Ababa, Jima, and

## 1. Introduction

Malaria disease is the leading health problem in Ethiopia [1, 2] because three-fourth (75%) of the total area of the country is malarious and more than two-third (approximately 68%) of the total population live below 2,000 m.a.s.l [3, 4]. However, the area coverage has not been revised in connection with changes such as urbanization, irrigation or dam, or land use and cover change [5].

Mizan-Tepi Universities had no role in the research design and data collection, conducting of research, and data analysis. Therefore, all authors declare no competing interest.

**Competing interests:** All authors declare no competing interest.

In Ethiopia, there are four types of *Plasmodium* species: *P. falciparum*, *P. vivax*, *P. malariae* and *P. ovale* [6], but *P. falciparum* and *P. vivax* are the most important parasites and found almost all parts of the country [3, 7]. These species are responsible for most of the burden of malaria (mortality and morbidity) in the country [8–10]. The proportion *P. falciparum* varied as compared with *P. vivax* in the country [8, 10–12] due to differences in altitude, rainfall, temperature, population movement, host and vector characteristics, and change in health care infrastructure [1, 13].

Recently, various reports in Ethiopia indicated that the burdens of malaria have shown decrement due to case management (nationwide implementation of ACT, artemisinin-based combination therapy), massive distribution of long lasting insecticidal treated nets (LLINs), and the application of indoor residual spraying (IRS) [14, 15]. Therefore, now the Ethiopian government has incorporated malaria elimination goal into its national strategy and allotted budget to support the goal [14].

In Ethiopia, over 42 species of *Anopheles* were recorded [16, 17], but the major malaria vector is *Anopheles arabiensis* [18, 19] while *An. pharoensis*, *An. funestus* and *An. nili* are secondary vectors [18, 20, 21]. These species are common in Bure district, Ethiopia, where this study was conducted [22].

The intensity of malaria transmission is measured by parous rates, daily survival rate, human blood meal frequency, sporozoite rate, and entomological inoculation rates [23–26]. Female parous status is a key index of vector competence, adult vector longevity, recruitment rate of adult, and the length of a gonotrophic cycle [27]. Of these, age and the ability of vectors to survive are among the most important factors in the epidemiology of vector-borne diseases [28], which depends on the proportion of mosquitoes that have ever fed on a human. Usually, older females also have higher exposure rates to malaria parasites during the previous human blood meal. Therefore, changes in the parous (already having laid at least one batch of eggs) rate reflects many aspects of the population change [27, 29].

The relation between the parous rate and the daily survival rate was given first by Davidson [30]. The longevity (the probability of daily survival) of the vector population is important for assessment of the efficacy of the vector control measures [31] which (longevity of adult *Anopheles*) varies between species and depends on external factors such as temperature, humidity and presence of predators [32]. In total, it is possible to calculate the probability of daily survival and to estimate the average longevity *Anopheles* mosquitoes using the proportion parous females and the duration of the gonotrophic cycle [33, 34]. However, in our study area, so far parous rates and the longevity of *Anopheles* mosquitoes have never been studied. Hence, the present study was aimed to investigate the parous rate, the daily survival and the longevity of *Anopheles* mosquitoes in Bure District, Northwestern Ethiopia.

## 2. Materials and methods

### 2.1. Study area

A longitudinal study was conducted in Bure district, northwestern Ethiopia, from July 2015 to June 2016. Geographically, Bure district is situated at an altitude ranging from 700 (Blue Nile gorge) to 2,350 m.a.s.l. Socioeconomically, the majority (85%) of the populations are farmers who grow maize, teff (*Eragrostis teff)*, pepper, potatoes, wheat, millets, followed by beans & peas, sunflower, niger, spices, vegetation's, and others; and the rest is merchants (6.8%) and others (non-governmental organizations, civil servants) (8.2%). Animals such as cattle, sheep, hens, mules, and donkeys are reared by the farmers. Additionally, both modern and traditional beekeepers were present. The majority of the population in the district live in houses made of mud and corrugated iron roofs. The mud-houses were partly smooth and partly rough. Some

of them were painted. The doors and windows of each house were lacking mosquito screening. In each farmer's compound, there was separated kitchen and latrine houses. Between each farmer's house, 10 -15m distance were present. Both animals and humans were living inside in the same houses (Personal obsr).

The majority of Bure districts has a subtropical zone (Woina-Dega) climate with annual mean minimum and maximum temperature of 9.9˚C and 29.2˚C, respectively and 2,000 mm mean annual rainfall range being 1,350–2,500 mm. The major rainy season of the district is from July to September, and a small amount is obtained from May to June and from October to December. The rest of the month (January—April) are dry seasons [35].

The study was conducted in three rural villages: Bukta, Workmidr and Shnebekuma, from July 2015—June 2016. The description of the detail of the three villages is found elsewhere [22]. Totally, these villages are malarious, bed nets were distributed to the three villages once per 3-years before malaria infestation begins, on the first week of September. Moreover, anti-malaria chemical spraying (IRS) (Deltamethrin, K-Othrine Flow) was administered to the three villages according to the national spraying operation guidelines [36].

## 2.2. Collection, identification, and processing of adult *Anopheles* mosquitoes

**2.2.1. Collection of adult *Anopheles* mosquitoes.** *Anopheles* mosquitoes were collected longitudinally from July 2015—June 2016. Entomological surveys were conducted monthly in each village, for one year using centers for disease control and prevention light trap catches (LTCs), pyrethrum spray catches (PSCs) and artificial pit- shelters (APSs). In each village, 9 houses for LTCs and 10 houses for PSCs were randomly selected and scattered in near to the breeding sites, in the middle and periphery sides of the village. In parallel, 27-LTs were pre-pared to collect the outdoor host seeking mosquitoes for the three villages, each had 9- LTCs. Additionally, six APSs were prepared in three villages to collect outdoor resting mosquitoes; each village had two. Techniques of indoor host-seeking and outdoor-resting mosquito collection were described elsewhere [22].

**2.2.2. Identification and processing of adult *Anopheles* mosquitoes.** Mosquitoes collected by LTCs, PSCs and APSs were identified morphologically at genus level using taxonomic keys mad by Verrone [37], Gillies and Coetzee [38], and Glick [39]. Morphologically identified and individually preserved *An. gambiae* specimens were identified by species-specific PCR as described by Wilkins *et al.* [40] at the molecular biology laboratory of tropical and infectious diseases research center, Jima University. Then, DNA was extracted from individual preserved *An. gambiae* complex species based on DNeasy Blood and Tissue Kits [41]. And then, DNA amplification was carried out. Following this, gel electrophoresis was carried out [40]. Finally, agarose-gel was placed on UVP. Those mosquitoes that remained unamplified were tested three times in an independent manner.

## 2.3. Determination of parous rate and longevity

Female *Anopheles* mosquitoes (*Anopheles arabiensis*, *An. funestus*, *An. coustani*, *An. squamosus*, *An. cinereus* and *An. pharoensis*) collected by LTs were analyzed for parous rate determination following the ovarian dissection. Dissection was carried out following protocol [42]. Ovaries were classified as parous (those *Anopheles* mosquitoes that have taken a blood meal at least once and laid eggs at least once and as a result the tracheoles have become stretched out) or nulliparous (*Anopheles* mosquitoes that have not taken a blood meal yet and have not laid eggs and as a result have females in which the ovaries have coiled tracheolar skeins) under a compound microscope using the 10x objective, and further confirmed using the 40x objective

[42]. Probability of daily survival was estimated as described in Ree *et al.* [33] and Ndoen *et al.* [34], whereas longevity of mosquito was determined as described in Davidson [30].

## 2.4. Data analysis

Data were entered and cleaned using 2007-microsoft excel and analyzed using SPSS software package version-20.0 (SPSS, Chicago, IL, USA). Parous rate, and longevity were calculated based on WHO [42]. Parous rate was estimated as the number of mosquitoes with parous ovaries divided by the number of females dissected multiplied by 100. The probability of survival of *Anopheles* species through one-day is equivalent to the cube root of the proportion of parous females in the population sample because there was no direct observation of the gonotrophic cycle (gc), the age estimation of these species was made on a 'gc' value of 3-days [30]. Therefore, the probability of surviving one day (denoted as p) for *Anopheles* species was estimated using the formula given by Ree *et al.* [33] and Ndoen *et al.* [34]:

$$P = \sqrt[3]{Proportion\ of\ Parous}$$

Whereas, mosquito life expectancy (longevity as d) was estimated following Ree *et al.* [33] and Ndoen *et al.* [34]:

$$d = \frac{1}{-lnP}$$

One way- ANOVA was applied to confirm the presence of parity difference in the villages ($p < 0.05$). Before running independent-samples T-test and ANOVA, normality of data was first checked and transformed [log10(x+1)]. When significant differences were observed in ANOVA, the Tukey test (HSD-Test) was used to separate the means ($p < 0.05$).

## 2.5. Ethics statement

A collection of mosquitoes was carried out after obtaining ethical approval from the ethics review committee of Addis Ababa University (reference no. CNSDO/382/07/15), Amhara Health Regional Bureau (permission reference no. H/M/TS/1/350/07) and the Head of the Bure District Health Office (permission reference no. BH/3/519L/2). Moreover, informed consent was obtained from the selected households.

# 3. Results

## 3.1. Composition and abundance of *Anopheles* mosquitoes

A total of 4,703 female *Anopheles* mosquitoes, representing nine species, were collected by all collection methods in Bure district. They were *Anopheles demeilloni* (50.7%), *An. arabiensis* (16.0%), *An. funestus* (13.6%), *An. coustani* (12.9%), *An. squamosus* (5.0%), *An. cinereus* (1.5%), *An. pharoensis* (0.32%), *An. rupicolus* (0.06%), and *An. natalensis* (0.02%). Of these species, *Anopheles demeilloni*, *An. arabiensis*, *An. funestus*, and *An. coustani* were found almost throughout the year. As compared to villages, the large proportion of these mosquitoes were collected in Shnebekuma than Bukta and Workmidr villages ($F_{2, 33} = 19.202$, $p < 0.0001$).

## 3.2. Parous rates of *Anopheles* mosquitoes

The results of ovary dissection for parous rate are shown in Tables 1–3 and Fig 1. A total of 952 unfed hosts-seeking *Anopheles* mosquitoes (*An. arabiensis*, n = 341; *An. funestus*, n = 239; *An. coustani*, n = 252; *An. squamosus*, n = 87; *An. cinereus*, n = 24; and *An. pharoensis*, n = 9) was dissected for parous rate determination. When compared, the overall parous rate was not

**Table 1. Parous rates and longevity of *Anopheles* mosquitoes caught by LTCs in Bure district, Ethiopia.**

| Mosquito Species | # Unfed | # Dissected | # Parous | % Parous | Daily Survival | Age (Days) |
|---|---|---|---|---|---|---|
| *An. arabiensis* | 498 | 341 | 177 | 51.9 | 0.804 | 4.6 |
| *An. pharoensis* | 9 | 9 | 4 | 44.4 | 0.763 | 3.7 |
| *An. funestus* | 382 | 239 | 151 | 63.2 | 0.858 | 6.5 |
| *An. coustani* | 436 | 252 | 106 | 42.1 | 0.749 | 3.5 |
| *An. squamosus* | 153 | 87 | 39 | 44.8 | 0.765 | 3.7 |
| *An. cinereus* | 41 | 24 | 6 | 25.0 | 0.630 | 2.2 |
| **All Total** | 1519 | 952 | 488 | 51.3 | 0.801 | 4.3 |

showing any statistically significant difference with nulliparous rate (t = 1.604; df = 10; p = 0.14). The parous rate of the six different species was compared, the result was different. The lowest parous rate was 25.0% for *An. cinereus* and the highest rate were 63.2% for *An. funestus*.

The overall parous rate of *An. arabiensis* in the district was 52.0%, range 35.0% (in Bukta) to 57.9% (in Shnebekuma) (Tables 1 and 2). The highest parous rate was recorded in Shnebekuma than other villages ($F_{2, 33}$ = 6.974; p = 0.003) (Table 3 and Fig 2). Seasonal variations in parous rate were observed in the district; the highest parous rate was recorded in March (69.0%) and the nil parous rate was observed in June (Fig 1).

The mean parous rate of *An. pharoensis* was 44.4%. This species was only collected in July and August months, and the high parous rate was 100% in July. The average parous rate of *An. funestus* was 63.2%. The peak parous rate was recorded in March (83.0%) and the lowest was in November (30%) for *An. funestus*. The overall parous rates for *An. squamosus*, *An. coustani*, and *An. cinereus* were 44.8%, 42.1%, and 25.0%, respectively. The parous rate of *An. cinereus* showed significant variation among villages ($F_{2, 33}$ = 5.044, p = 0.012) and the highest rate (63.0%) was recorded in Bukta. There was variation in monthly mean parous rates of three species (*An. coustani*, *An. squamosus* and *An. cinereus*) and the peak parous rate was recorded in October (94.0%), in October (73.0%) and in May 100.0%), respectively (Fig 2).

### 3.3. Longevity of *Anopheles* mosquitoes

Tables 1 and 2 show the daily survival and longevity of different *Anopheles* species. Age variation was observed between and within species in the study villages. The mean life expectancy of *An. funestus* was 6.5 days, which was higher than other species. The mean longevity of *An. arabiensis*, *An. coustani*, *An. squamosus*, *An. pharoensis*, and *An. cinereus* was 4.6 days, 3.5 days, 3.7 days, 2.7 days, and 2.2 days, respectively (Table 1). Except for *An. coustani* (L = 4.7), the rest species had shown the highest longevity in Shnebekuma than other villages (Table 2).

**Table 2. Parous rates and longevity of *Anopheles* mosquitoes by villages caught by LTCs in Bure district, Ethiopia.**

| Species | Bukta | | | | Workmidr | | | | Shnebekuma | | | |
|---|---|---|---|---|---|---|---|---|---|---|---|---|
| | D | %p | DS | L | D | %P | DS | L | D | % | DS | L |
| *An. arabiensis* | 52 | 34.6 | 0.70 | 2.8 | 49 | 40.8 | 0.74 | 3.4 | 240 | 57.9 | 0.83 | 5.5 |
| *An. pharoensis* | 4 | 50.0 | 0.79 | 4.3 | 0 | 0 | 0 | 0 | 3 | 66.7 | 0.87 | 7.4 |
| *An. funestus* | 15 | 33.3 | 0.69 | 2.7 | 16 | 25.0 | 0.63 | 2.2 | 142 | 67.9 | 0.88 | 7.6 |
| *An. coustani* | 40 | 27.5 | 0.65 | 2.3 | 67 | 52.2 | 0.81 | 4.7 | 145 | 41.4 | 0.75 | 3.4 |
| *An. squamosus* | 6 | 33.3 | 0.69 | 2.7 | 13 | 30.8 | 0.68 | 2.6 | 70 | 47.1 | 0.78 | 4 |
| *An. cinereus* | 22 | 27.3 | 0.65 | 2.3 | 2 | 0 | 0 | 0 | 0 | 0 | 0 | 0 |

Note: D = No of dissected; P = Parity; DS = Daily survival; L = longevity.

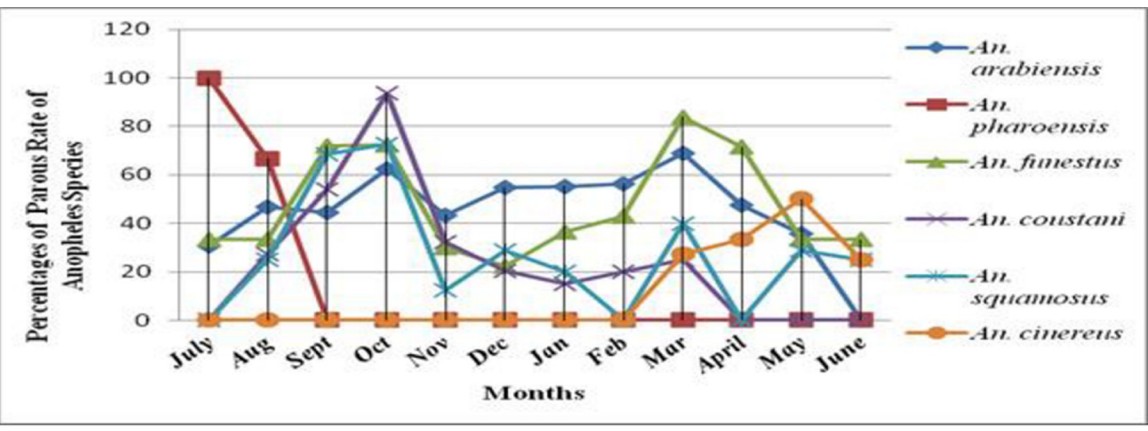

**Fig 1. Mean monthly parous rates of *Anopheles* mosquito in Bure district, Ethiopia (July 2015—June 2016).**

## 4. Discussion

Malaria is transmitted by human blood-feeding *Anopheles* mosquitoes. In our study, the well-known primary (*An. arabiensis*) and secondary malaria vectors (*An. funestus* and *An. pharoensis*) of Ethiopia were identified [14, 36]. Various studies conducted in Ethiopia also reported the occurrence of these species in the higher altitudes (ranged 2010–2280 m.a.s.l) [43–45]. *An. arabiensis* and *An. funestus* were found throughout the year [22] and were found to be infected by malaria parasites [46].

The parous rate relates not only to the daily survival rate of adults, but also to the recruitment rate of adults, the adult longevity and the length of a gonotrophic cycle. Therefore,

**Table 3. Mean parous rate of *Anopheles* mosquito in the villages, Bure district, Ethiopia.**

| Villages | Species | M ± se | F$_{2, 33}$ | p- value |
|---|---|---|---|---|
| Bukta | *An. arabiensis* | 0.0713 ± .020[b] | 6.974 | 0.003 |
| Wormidr | | 0.102 ± .023[b] | | |
| Shnebekuma | | 0.179 ± .020[a] | | |
| Bukta | *An. pharoensis* | 0.0502 ± .034 | 1.094 | 0.347 |
| Wormidr | | 0.000 ± .000 | | |
| Shnebekuma | | 0.0398 ± .028 | | |
| Bukta | *An. funestus* | 0.052 ±.019 | 2.75 | 0.079 |
| Wormidr | | 0.033 ±.018 | | |
| Shnebekuma | | 0.371 ±.197 | | |
| Bukta | *An. coustani* | 0.059±.079 | 0.143 | 0.867 |
| Wormidr | | 0.058±.089 | | |
| Shnebekuma | | 0.076±.101 | | |
| Bukta | *An. squamosus* | 0.029±.019 | 2.284 | 0.118 |
| Wormidr | | 0.024±.017 | | |
| Shnebekuma | | 0.084±.028 | | |
| Bukta | *An. cinereus* | 0.042±.0187[a] | 5.044 | 0.012 |
| Wormidr | | 0.000± 0.00[b] | | |
| Shnebekuma | | 0.000± 0.00[b] | | |

Note: Mean (s) followed by the same letter (s) in the same column are not significantly different from each other at p < 0.05 (Tukey HSD).

M = mean, se = standard error.

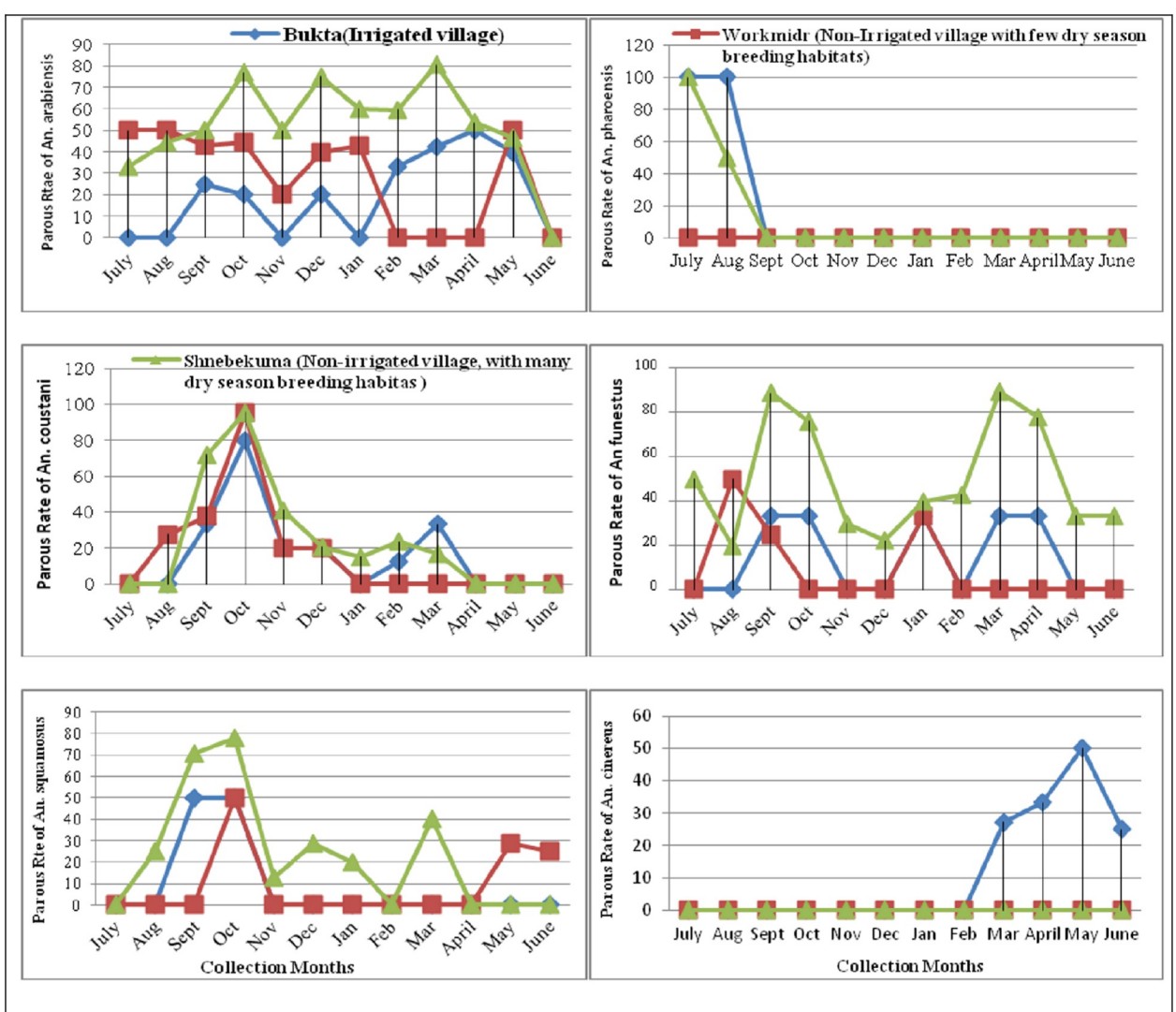

**Fig 2. Mean parous rates of *Anopheles* mosquito in three villages, Bure district, Ethiopia (July 2015—June 2016).**

changes in the parous rate reflect many aspects of the population changes [27]. In this study, the dissection of five *Anopheles* species was examined to determine the parous rate and the daily survival of the five species. The mean parous rate of *An. arabiensis* in the district was 52.0%. It is in agreement with the parous rate of *An. gambiae* s.l (49.3%) in southeastern Tanzania [47], but higher than a study conducted in East Wellega Zone, Ethiopia with an overall *An. arabiensis* parous rate was 45% [48]. On the other hand, it was lower as compared with the parous rate *An. arabiensis* in Sille (73.2%) [49], in Lare District (63%) [50], and in Metema-Armachiho lowland (69.4%) [51], Ethiopia, and in a Peri-urban area of plateau state (77.8%), Northcentral Nigeria [52]. The higher proportion of parous vectors may also be a consequence of the suitable climate conditions in the area, high relative humidity which is connected with the presence highly extended marshlands which largely influences the mosquito biology [53]. The mean annual parous rates in our study sites suggested that this species had already practiced haematophagy. Seasonally, the highest parous rate for this species was recorded in March (dry month) and zero parous rate was observed in June (pre-rainy season). This is probably

due to the presence of sufficient numbers of mature adults in March; on the other hand, the absence of in June could be associated with the proper usage of LLINs. Being this practice, this species probably was deprived of any blood sources or it could be the presence of nulliparous mosquitoes only.

As compared to villages, the higher parous rate for *An. arabiensis* was recorded in Shnebekuma than other villages. This difference could be probability associated with the presence of highly extended marshlands that can release sufficient humidity to the surround area as compared with other villages [54]. In general, the recorded mean monthly parous rate of *An. arabiensis* in our study indicates that this vector is not long lived and less efficient to transmit malaria, though *Plasmodium* infected *An. arabiensis* was obtained in the Bure district [46].

The mean parous rate of *An. pharoensis* was 33.0%, which is disagreement with Taye *et al*. [50] finding, who reported 55% mean parous rate of this species in Lare District, Ethiopia. The present surveyed area has an elevation between 2,024 to 2,157m.a.s.l., which is relatively highland (cold) and not comfortable for the adult survival. Being this, the density of the adult of *An. pharoensis* was very low throughout the year [22]. Because, naturally *An. pharoensis* is a major transmitter in arid and semiarid regions [38, 55]. This species was only trapped in July and August and the highest parous rate was seen in July, through July and August are the main rainy season in Ethiopia [35]. The absence of any parous rate of *An. pharoensis* in most months could be connected with low temperature.

The mean parous rate of *An. funestus* in the study area was 63.0%, which was comparable with Kaindoa *et al*. [47] finding, who reported 65.8% mean parous rate for *An. funestus* in southeastern Tanzania. However, it is inconsistent with Tanga *et al*. [56] finding, who reported 82.0% mean monthly parous rate at 281 m.a.s.l in Cameroon. The reason for such various may be the presence of altitudinal difference (temperature) [53]. Additionally, in this study, the highest parous rate *An. funestus* was observed in March (83.0%). This observation could be connected with the trapping of large numbers of mature *An. funestus*, which were supported by the presence of extended marshlands (sufficient humidity) [54] and various blood meals sources in most surveyed houses in the study area [57].

Moreover, the overall parous rates for *An. coustani*, *An. squamosus* and *An. cinereus* were 42.0%, 45.0% and 25.0%, respectively. These parous rates for these species indicated that they are not long lived to complete the parasite life cycle within, though *Plasmodium* infected *An. coustani* was detected in the study area [46]. This *Plasmodium* infected species had shown peak parous rate in October (94.0%), due to the presence of enough blood meals [58] combined with the occurrence of enough atmospheric humidity, which elongate the life of *Anopheles* mosquitoes [59–61].

Vectorial role of mosquitoes in disease transmission (malaria) is influenced by life expectancy (longevity) [34]. In this study, the overall longevity of *An. arabiensis* (4.5 days) is not comparable with the findings of Gari *et al*. [10] and Taye *et al*. [50, 62], who documented a longevity between 7 to 25 days, and 5.3 to 6.9 days of *An. arabiensis* (*An. gambiae* s.l) in different parts of Ethiopia. Therefore, in the study district the average longevity of *An. arabiensis* was not sufficient to complete the life cycle of malaria parasite for malaria transmission throughout the year because *P. falciparum* requires from 12–14 day at 25˚C and between 22–23 at 20˚C. Also, *P. vivax* requires 9–10 at 25˚C and 16–17 days at 25˚C to complete their life cycle in mosquitoes [31]. Similarly, Ndoen *et al*. [34] noted that the average duration life of female *Anopheles* in tropical areas is about 10–14 days. Thus, in our research, the longevity of this species is insufficient for pathogen development.

In the current study, the overall longevity of *An. pharoensis* was 2.7 days (ranged 0–7.4), which is longer than 1.6 days (ranged from 0–1.8 days) reported in Ethiopia [10]. On the other hand, Taye *et al*. [50] reported 5.9 average days (ranged from 3.3–8.6 days) of *An. pharoensis*

in Ethiopia. The low life expectancy of *An. pharoensis* in this study may be due to the collection of a small proportion of the population of *An. pharoensis*, which was only trapped from July to September [22]. Totally, the average longevity of *An. pharoensis*, *An. funestus*, *An. coustani*, *An. squamosus* and *An. cinereus* were very low for the completion of the sporogonic cycle in their bodies [31, 34].

## 5. Conclusions

The current study identified nine *Anopheles* mosquitoes, of which *An. arabiensis*, *An. funestus* and *An. pharoensis* were the most important malaria vector of Ethiopia. The parous rate of the six different species was determined, the highest annual rate was 63.2% for *An. funestus*. In total, the annual parous rate of *An. arabiensis*, *An. funestus* and *An. pharoensis*, *An. coustani*, *An. squamosus* and *An. cinereus* were very minimum. Similarly, the average longevity of these species was not sufficient to complete the life cycle of malaria parasite.

## Supporting information

**S1 File.**
(DOCX)

## Acknowledgments

We thank the communities of Bukta, Shnebekuma, and Workmidr, Bure district, for their cooperation and protection of the CDC-LTs during our field survey. Our deepest gratitude goes to Malaria Consortium, Ethiopia, for providing CDC LTs and internet services. We acknowledge to Dr. Habte Tekie for his appreciation and provision of CDC LT lamps (lightbulbs).

## Author Contributions

**Conceptualization:** Tilahun Adugna, Emana Getu, Delenasaw Yewhelew.

**Data curation:** Tilahun Adugna.

**Formal analysis:** Tilahun Adugna.

**Investigation:** Tilahun Adugna.

**Methodology:** Tilahun Adugna, Delenasaw Yewhelew.

**Resources:** Emana Getu.

**Software:** Tilahun Adugna.

**Supervision:** Emana Getu.

**Visualization:** Delenasaw Yewhelew.

**Writing – original draft:** Tilahun Adugna.

**Writing – review & editing:** Tilahun Adugna, Emana Getu, Delenasaw Yewhelew.

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
