## [Editor Report · Decision Letter 0]

9 Oct 2021

PONE-D-21-21932Parous rate and longevity of anophelines mosquitoes in Bure district, northwestern EthiopiaPLOS ONE

Dear Dr. Adugna,

Thank you for submitting your manuscript to PLOS ONE. After careful consideration, we feel that it has merit but does not fully meet PLOS ONE’s publication criteria as it currently stands. Therefore, we invite you to submit a revised version of the manuscript that addresses the points raised during the review process.

We look forward to receiving your revised manuscript.

Kind regards,

Clive Shiff

Academic Editor

PLOS ONE

Journal Requirements:

2. Thank you for including the following ethics statement on the submission details page:

'A collection of mosquitoes was carried out after obtaining ethical approval

from the ethics review committee of Addis Ababa University (reference no.

CNSDO/382/07/15), Amhara Health Regional Bureau (permission reference no.

H/M/TS/1/350/07) and the Head of the Bure District Health Office (permission

reference no. BH/3/519L/2). Moreover, informed consent was obtained from

the selected households.'

Please also include this information in the ethics statement in the Methods section of your manuscript.

Additional Editor Comments (if provided):

I like this paper, you have written it well and it should be published. My concern, and I would like you to rewrite this part. This deals with the introduction and the discussion of parous rates, Parity or non-parous are two conditions. A parous mosquito could have laid one or more batches of eggs. Th only way to expand parity is via Detinova's technique. But it is very difficult to undertake and be accurate. I know I have done it. You quote from ref 27about the length of a gonotrophic cycle, i.e. the time passed between laying egg batches. Really this can only be done in caged mosquitoes, but as they are poikilothermic, the ambient temperature is important. I have read ref 27, it is somewhat hypothetical, and I am not sure it is feasible to compare with nature.

We all have tried to age the life span of female mosquitoes, but the answer is frustratingly difficult.

Please also review the sentence "Usually, older females also have higher exposure rates... it is unclear. please do not suggest that we can judge the life span of a mosquito by parous rates, because we cannot estimate how many ovipositions had taken place. please also note that Davidson's work on longevity has been disputed, (see ref 27)
---

## [Author Response · Author response to Decision Letter 0]

7 Nov 2021

Response letter

Comment 1: Prepare based on PLOS ONE journal guideline

Response 1: We have tried to prepare the manuscript based on the PLOS ONE journal guideline 

 criteria. 

Comment 2: Regarding “ethical statement” 

Response 2: As indicated by the reviewer / editor, ethical statements are included under materials 

 and methods section. In particular, next to data analysis. It is highlighted in the 

 manuscript. 

Comment 3: About “Funding Information’ and ‘Financial Disclosure” 

Response 3: We didn’t get any fund source from any organization. Being this, we clearly indicated 

 in the that “The author(s) received no specific funding for this work” in the funding 

 information section.

Comment 4. About data access / provision. 

Response 4: I corrected myself. It is possible to access any necessary data /no restriction/ if you 

 require. Being this, revision of the cover letter is not necessary because there are 

 no data restrictions. All necessary data for this manuscript is already uploaded. If 

 the reviewer is required raw data; I can do now. 

Comment 5: About ethical statement/ repeated/

Response 5: Enough information about ethical statements is treated under material method 

 section. All available and required information are incorporated. 

Comment 6: Regarding references lists, i.e., review your reference list to ensure that it is 

 complete and correct.

Response 6: All used references are cited in the manuscript and follow the correct ways of 

 referencing style. There is no new citation and reference add in the manuscript. 

Additional Editor Comments 

Comment: Please also review the sentence "Usually, older females also have higher 

 exposure rates..................It is indicated in the introduction. 

Response: If you see the truth, naturally female known by feeding blood meals from 

 various animals including human to mature the fertilized eggs. Until she gets 

 die, she will have many batches of eggs. For each batches, she will take 2-3 

 blood meals. Therefore, to have many batches of eggs, the female should take 

 many times of blood meals. This increase the rate of exposure to be infected by 

 the parasites. Because of having this kinds of truth, we prefer to use the phrase 

 “usually, older females also have higher exposure rates….”.

 In conclusion, the phrase hasn’t any problem /it is science/

Comment: About calculating the life expectancy of mosquitoes

 Response: Partly, I agree with your saying, but we have no any other option to do it. In 

 similar fashion (objective), the following researchers have used the formal 

 driven by Davidson (1954) to calculate mosquito life expectancy (longevity). 

[34] Ndoen, E., Wild, C., Dale, P., Sipe, N., Dale, M. (2012). Mosquito Longevity, Vector 

 Capacity, and Malaria Incidence in West Timor and Central Java, Indonesia. 

 International Scholarly Research Network. doi:10.5402/2012/143863.

[62] Taye B, Lelisa K, Emana D, Asale A, Yewhalaw D. Seasonal dynamics, longevity and 

 biting activity of anopheline mosquitoes in southwest Ethiopia. J Insect Sci. 

 2016;16:1-7. doi: 10.1093/jisesa/iev150 

 Therefore, we can say that we are correct and haven’t any choice.

Comments: To prepare figures using PACE

Response: I did it after I registered. The uploaded figures are developed using the given 

 comments. It meets the requirement of PLOS ONE criteria, based on PACE.

---

## [Editor Report · Decision Letter 1]

17 Jan 2022

Parous rate and longevity of anophelines mosquitoes in Bure district, northwestern Ethiopia

PONE-D-21-21932R1

Dear Dr. Adugna,

We’re pleased to inform you that your manuscript has been judged scientifically suitable for publication and will be formally accepted for publication once it meets all outstanding technical requirements.

Kind regards,

Clive Shiff

Academic Editor

PLOS ONE
---

## [Editor Report · Acceptance letter]

26 Jan 2022

PONE-D-21-21932R1 

*Parous rate and longevity of anophelines mosquitoes in bure district, northwestern Ethiopia*

Dear Dr. Adugna:

I'm pleased to inform you that your manuscript has been deemed suitable for publication in PLOS ONE. Congratulations! Your manuscript is now with our production department. 

Kind regards, 

on behalf of

Dr. Clive Shiff 

Academic Editor

PLOS ONE